# Clinical Practice Guidelines for Therapeutic Drug Monitoring of Vancomycin in the Framework of Model-Informed Precision Dosing: A Consensus Review by the Japanese Society of Chemotherapy and the Japanese Society of Therapeutic Drug Monitoring

**DOI:** 10.3390/pharmaceutics14030489

**Published:** 2022-02-23

**Authors:** Kazuaki Matsumoto, Kazutaka Oda, Kensuke Shoji, Yuki Hanai, Yoshiko Takahashi, Satoshi Fujii, Yukihiro Hamada, Toshimi Kimura, Toshihiko Mayumi, Takashi Ueda, Kazuhiko Nakajima, Yoshio Takesue

**Affiliations:** 1Division of Pharmacodynamics, Faculty of Pharmacy, Keio University, Tokyo 105-8512, Japan; matsumoto-kz@pha.keio.ac.jp; 2Department of Pharmacy, Kumamoto University Hospital, Kumamoto 860-8556, Japan; kazutakaoda@kuh.kumamoto-u.ac.jp; 3Division of Infectious Diseases, Department of Medical Subspecialties, National Center for Child Health and Development, Tokyo 157-8535, Japan; shoji-k@ncchd.go.jp; 4Department of Pharmacy, Toho University Omori Medical Center, Tokyo 143-8541, Japan; yuuki.hanai@med.toho-u.ac.jp; 5Department of Pharmacy, Hyogo College of Medicine, Nishinomiya 663-8501, Japan; yktabu@hyo-med.ac.jp; 6Department of Hospital Pharmacy, Sapporo Medical University Hospital, Sapporo 060-8543, Japan; fujii.satoshi@sapmed.ac.jp; 7Department of Pharmacy, Tokyo Women’s Medical University Hospital, Tokyo 162-0054, Japan; hamada.yukihiro@twmu.ac.jp (Y.H.); kimura.toshimi@twmu.ac.jp (T.K.); 8Department of Emergency Medicine, School of Medicine, University of Occupational and Environmental Health, Fukuoka 807-8555, Japan; mtoshi@med.uoeh-u.ac.jp; 9Department of Infection Prevention and Control, Hyogo College of Medicine, Nishinomiya 663-8501, Japan; taka76@hyo-med.ac.jp (T.U.); nakajima@hyo-med.ac.jp (K.N.); 10Department of Clinical Infectious Diseases, Tokoname City Hospital, Tokoname 479-8510, Japan

**Keywords:** model-informed precision dosing, vancomycin, therapeutic drug monitoring, area under the concentration-time curve, guideline

## Abstract

Background: To promote model-informed precision dosing (MIPD) for vancomycin (VCM), we developed statements for therapeutic drug monitoring (TDM). Methods: Ten clinical questions were selected. The committee conducted a systematic review and meta-analysis as well as clinical studies to establish recommendations for area under the concentration-time curve (AUC)-guided dosing. Results: AUC-guided dosing tended to more strongly decrease the risk of acute kidney injury (AKI) than trough-guided dosing, and a lower risk of treatment failure was demonstrated for higher AUC/minimum inhibitory concentration (MIC) ratios (cut-off of 400). Higher AUCs (cut-off of 600 μg·h/mL) significantly increased the risk of AKI. Although Bayesian estimation with two-point measurement was recommended, the trough concentration alone may be used in patients with mild infections in whom VCM was administered with q12h. To increase the concentration on days 1–2, the routine use of a loading dose is required. TDM on day 2 before steady state is reached should be considered to optimize the dose in patients with serious infections and a high risk of AKI. Conclusions: These VCM TDM guidelines provide recommendations based on MIPD to increase treatment response while preventing adverse effects.

## 1. Introduction

Vancomycin has been widely used to treat infections caused by methicillin-resistant *Staphylococcus aureus* (MRSA) [1,2,3,4,5]. Vancomycin treatment is associated with several adverse effects such as acute kidney injury (AKI) [6,7,8] and ototoxicity [9,10]. Because of the narrow therapeutic range of vancomycin, therapeutic drug monitoring (TDM) is mandatory for maximizing efficacy while preventing these adverse effects. A committee consisting of the Japanese Society of Chemotherapy (JSC) and the Japanese Society of Therapeutic Drug Monitoring (JSTDM) developed Japanese practice guidelines for TDM of vancomycin in 2013 [11]. Ye et al. conducted a systematic review of 12 guidelines for vancomycin TDM using the Appraisal of Guidelines for Research & Evaluation II [12], and a consensus guideline developed by the American Society of Health-System Pharmacists, the Infectious Diseases Society of America (IDSA), and the Society of Infectious Diseases Pharmacists [13] and the Japanese guideline had the highest scores in this domain, and only these two guidelines were recommended.

At that time, trough concentrations of 15–20 μg/mL were recommended as the surrogate of AUC/MIC > 400 excluding MIC = 2 μg/mL for the treatment of complicated infections [13]. By contrast, because of the insufficient data supporting the safety of sustained trough concentrations of 15–20 μg/mL, the previous Japanese guideline recommended that initial therapy should be started at a dosage sufficient to achieve trough concentrations of 10–15 μg/mL even in patients with complicated infections, and the dose can be adjusted after TDM to achieve trough concentrations of 15–20 μg/mL according to the treatment response [11].

In the era in which model-informed precision dosing (MIPD) became a possible tool in clinical practice, two guidelines for TDM of vancomycin have been recently published [3,14]. The revised American guideline reported by Rybak et al. recommended AUC-guided dosing, and trough-guided dosing targeting 15–20 μg/mL was no longer recommended for serious MRSA infections [3]. Conversely, the updated Chinese guideline recommended trough-guided dosing and AUC-guided dosing at the same strength, considering institutions in which Bayesian software is not available and the possibility of poor feasibility of measuring two-point samples in a first-order pharmacokinetic (PK) equation [13]. In contrast, these guidelines aimed to provide recommendations emphasising the importance of the MIPD approach to TDM practice for vancomycin. The committee conducted several studies including a systematic review and meta-analysis for evidence-based recommendations.

## 2. Materials and Methods

### 2.1. Preparing the Guideline

The clinical practice guideline committee was consisted of 18 experts from the JSC and JSTDM. The committee reviewed the previous Japanese clinical practice guidelines for TDM of vancomycin [11]. The updated guidelines were developed according to ‘Medical Information Network Distribution Service (Minds) Manual for Guideline Development 2017′ in Japan [15]. The committee selected the 10 clinical questions (CQs), and identified original articles related to these CQs through general databases (i.e., MEDLINE, Web of Science, EMBASE, Cochrane Library).

Systematic reviews of the CQs were performed by committee members to make recommendations based on current evidences. Because randomized controlled trials (RCTs) on the practice of TDM is difficult to perform, ranking recommendations and evidence levels adopted in the guidelines were made using the modified Minds classification (Table 1).

### 2.2. Systematic Review and Meta-Analysis (for CQs 2 and 6)

We searched three electronic databases (MEDLINE, Web of Science, and Cochrane Central Register of Controlled Trials) for clinical studies published through 9 August 2020 using a text search strategy with the following terms: “vancomycin”, “trough”, and “monitoring” for the evaluation of VCM trough concentrations; “vancomycin”, “AUC”, or “area under the curve” for the evaluation of VCM target AUC values; and “vancomycin” or “monitoring” for the evaluation of different monitoring strategies [17]. We excluded studies that met the following criteria: questionnaire study, letter, case report, and review articles; non-adult patients or non-human subjects; the VCM dose was not adjusted using the AUC or trough; and detailed results were not available in English for the evaluation of different monitoring strategies. After removing duplicates, our initial search returned 3293 studies for the evaluation of VCM trough concentrations, 1029 studies for the evaluation of VCM target AUC values, and 3156 studies for the evaluation of different monitoring strategies. After screening the titles and abstracts, the full text of the remaining 86, 31, and 6 articles was assessed in detail.

For the evaluation of different PK monitoring strategies, two studies assessing effectiveness (*n* = 326) [18,19] and four studies assessing safety (*n* = 1902) [18,19,20,21] were selected for meta-analyses. For the evaluation of vancomycin trough concentrations, effectiveness was assessed at a cut-off of 10 (two studies, *n* = 393) [22,23] or 15 µg/mL (eight studies, *n* = 849) [22,23,24,25,26,27,28,29], and safety was assessed through a comparison of 10–15 and <10 µg/mL (14 studies, *n* = 1637) [22,30,31,32,33,34,35,36,37,38,39,40,41], through a comparison of 15–20 and 10–15 µg/mL (12 studies, *n* = 1211) [22,30,31,33,34,36,37,38,39,40,41,42], and through a comparison of ≥20 and 15–20 µg/mL (14 studies, *n* = 1127) [22,30,31,33,34,36,37,38,39,40,41,42,43,44]. For the evaluation of vancomycin target AUC values, five studies assessing effectiveness (*n* = 383) [23,27,45,46,47] at an AUC/MIC cut-off of 400 ± 15 (392.7–451) and five studies assessing safety (*n* = 1305) [20,48,49,50,51] at an AUC cut-off of 600 ± 15 (550–683) were selected for meta-analyses.

### 2.3. Studies Conducted by the Committee to Make Recommendations for Each CQ

Candidates for AUC-guided dosing (for CQ 3): A multicentre retrospective cohort study was conducted by the committee to identify risk factors for vancomycin-induced AKI and candidates for AUC-guided dosing, who require a more accurate dose titration to reduce the risk of AKI [52]. We obtained healthcare data retrospectively from the electronic medical records of seven acute-care university hospitals to which the committee members are affiliated. Of the 3759 eligible patients, 1877 were excluded, leaving 1882 patients included in the evaluation.

Development of software to distribute AUC-guided dosing (for CQs 1 and 4): We newly developed practical AUC-guided therapeutic drug monitoring (PAT) for AUC estimation. PAT was developed on the R ver. 3.6.2 platform for use on mobile phones and personal computers (https://pharmacokinetic-simulation.shinyapps.io/app-ver1/; accessed on 16 February 2022) [53]. AUC estimated by PAT using the trough level alone and trough and peak measurements was evaluated against the reference AUC calculated with the log-linear trapezoidal rule using eight measured concentrations.

Timing of TDM (for CQ 5): AUC on days 1 and day 2 and at steady state was evaluated using PAT in 260 patients with MRSA infections, and the relationships of AUC on day 2 with early treatment response and AKI were studied [54].

Loading dose (for CQ 8): A single-centre retrospective cohort study was conducted by the committee to evaluate whether a loading dose improved early clinical response without increasing the risk of AKI [55]. Overall, 157 patients were included in the safety analyses, and 82 patients were analysed for early clinical responses.

The regimen to achieve AUC targets (for CQ 9): A retrospective study was conducted between April 2011 and May 2020 to identify the regimen that increased the probability of achieving AUC targets. The study included adult patients with an estimated glomerular filtration rates (eGFRs) of ≥70 mL/min/1.73 m^2^. Four different regimens were evaluated (A-1 [*n* = 69], A-2 [*n* = 20], B-1 [*n* = 64], and B-2 [*n* = 71]). A loading dose was used in regimens B-1 (25 mg/kg) and B-2 (30 mg/kg). The maintenance dose was 15 mg/kg twice daily in regimens A-1 and B-1 and 20 mg/kg twice daily in regimens A-2 and B-2. Dose rounding within 2.5 mg/kg was permitted for the analysis. The exact sampling time after the previous dose was used to calculate AUC using Bayesian software PAT [53].

The regimen for achieving AUC targets in patients with impaired kidney function (for CQ 10): The committee recommended Bayesian software for optimal dosing in patients with impaired kidney function. To obtain a tentative regimen to input into the computer, the committee proposed a nomogram using the mean population PK parameter.

### 2.4. Process before Publication

External public comments for the draft guidelines which were uploaded to the home pages of JSC and JSTDM were obtained between 2 June 2021 and 2 July 2021. The Japanese version of the guidelines was approved by the JSC and JSTDM Board of Directors, and the summary was published in *Japanese Journal of Chemotherapy* in February 2022. In this international version, we further revised these guidelines, particularly focusing on the importance of MIPD. Potential conflicts of interest are listed in the Acknowledgments section. The committee planned revision of the guidelines at 5-year intervals.

## 3. Results

### 3.1. Executive Summary


*CQ 1. How can MIPD software be used to increase the accuracy of dose individualization?*


The category of recommendations and evidence levels were not demonstrated in this CQ. CQ 1 was selected to better understand the recommendations described in CQs 2–10.

1)Because vancomycin has a narrow therapeutic window, maintaining exposure for better treatment efficacy and less toxicity is essential for antimicrobial stewardship [3,17,56]. MIPD applies a concept for interpreting drug concentrations including PK calculations along with significant covariates [57,58,59,60].2)Population PK models can serve as a quantitative PK framework in MIPD.
a)Population PK models are used to guide initial therapeutic decision making (population PK model-guided dosing) [54,58,61,62,63].b)Population PK model-guided dosing is optimised to dosing based on the Bayesian posterior probability using the observed individual patient PK/pharmacodynamic (PD) information (Bayesian estimation using the measured concentration combined with a population PK model) [56,64,65].c)Better prior probability was reported in a population PK model based on rich sampling (full data set: e.g., at the end of infusion, at 60, 120, and 300 min following the infusion, and immediately before the next dose) than that based on limited sampling (e.g., trough and peak concentrations) [66,67,68,69,70,71].d)Population properties (i.e., age, body weight, kidney function, other potential covariates) for establishing a population PK model should be considered to determine the reasonable candidates for the MIPD software to increase the accuracy of dosing [53,72,73,74,75,76,77,78,79,80,81,82,83,84,85,86,87,88,89,90,91,92,93,94,95,96,97].e)The Bayesian prior information has been accumulated in special populations of obesity, paediatrics, and renal replacement therapy (RRT).3)MIPD tools can integrate this complex information to help individualize the rational dosing of vancomycin [53,98,99,100,101].4)Compared to traditional TDM, the MIPD approach has several advantages to streamline the TDM process of vancomycin.
a)MIPD offers prompt AUC calculation for the management of MRSA infections, in which AUC/MIC is recommended for use as the PK/PD target [3,17,57,58,59,60].b)MIPD proposes AUCs on days 1 and 2 and at steady state irrespective of the day of TDM, and the antimicrobial stewardship team can select any of these AUCs to achieve the target range according to their needs [18,54].c)MIPD does not require steady state to be reached, and earlier concentration data are available than obtained using traditional TDM for the adjustment of vancomycin dosing.d)MIPD offers an opportunity to handle concentrations measured at any time during the treatment course, allowing for more flexibility in the timing of sampling.e)MIPD utilizes a patient’s entire dosing information including the loading dose and concentrations in calculations during treatment, and cumulative data are included for estimating AUC in patients with modified dosing regimens because of the TDM process.


*CQ 2. What are the recommended PK/PD parameters for TDM?*


1)AUC/MIC (I) and AUC (III-A) indicate efficacy and safety, respectively [3,11,102,103,104,105,106,107,108,109,110].2)Trough levels should not be substituted for AUC [56,66,111,112,113] (IV).3)Although a target trough level of 15–20 μg/mL has been recommended for clinical efficacy in severe/complex MRSA infection, it poses a risk of AKI [11,17,52,110] (IV).4)AUC-guided dosing is more strongly recommended to decrease the risk of AKI than trough-guided dosing [17,19,21,114,115,116] (III-A).5)AUC-guided dosing may be more clinically effective than trough-guided dosing, although relevant studies are limited [17,19,114] (III-B).


*CQ 3. Who are the candidates for AUC-guided dosing?*


1)When feasible, the routine use of AUC-guided dosing is suggested irrespective of the severity or complexity of MRSA infections because it decreases the risk of AKI (III-A).2)Even in institutions in which routine use of the Bayesian approach is difficult, the introduction of AUC-guided dosing should be considered for patients at high risk of AKI because of impaired kidney function, concomitant use of piperacillin/tazobactam (PIPC/TAZ) or diuretics, and intensive care unit (ICU) stay [6,17,52,117,118,119,120,121,122,123,124] (II).3)Patients with altered V_d_ and renal function and those with unstable haemodynamics might be candidates for AUC-guided dosing to increase the prediction accuracy [42,52,96,117,118,119,125,126,127,128,129,130,131,132,133,134,135,136,137,138,139,140] (III-A).


*CQ 4. Is the trough concentration alone sufficient to estimate AUC using Bayesian software?*


1)Because most published population PK models were based on limited sampling, two-point measurement (trough level and peak level at 1–2 h after infusion) is recommended, especially in patients with impaired kidney function who receive vancomycin over a 24-h interval, patients at risk of vancomycin-induced AKI, and those with serious or complicated MRSA infections [42,52,53,54,56,63,66,96,110,112,118,125,126,127,130,137,140] (II).2)AUC estimated using trough-only data is more reliable when Bayesian software based on a population PK model with rich sampling is used [66] (II).3)Trough-only measurements may be used for Bayesian estimation in patients with mild infections who received vancomycin q12h [53] (III-A).4)Although it is recommended to measure the trough level within 30 min before dosing, its measurement timing tends to be incorrect in actual clinical practice [18]. However, blood samples can be taken at random times in Bayesian estimation (III-A), and the exact times before (or after) dosing should be used in Bayesian estimation.


*CQ 5. When should TDM be performed?*


1)Using the MIPD approach, TDM on day 2 before steady state is reached should be considered in patients with serious or complicated MRSA infections, patients at risk of AKI, and those with unstable renal function [26,51,54,141] (III-A). With dose optimisation based on TDM in the morning on day 2, the probability of achieving day 2 AUC (24–48 h) targets can be increased.2)Candidates for day 2 TDM and two-point measurement have similar patient/infection characteristics (e.g., high risk of AKI or serious infection). Therefore, two-point measurement (e.g., before [trough] and 1–2 h after [peak] the third dose) is a reasonable approach in ICU patients in whom vancomycin is administered q12h and TDM is performed on day 2 (III-A).3)As mentioned in CQ 4, two-point measurement was recommended in patients receiving q24h administration. It should be considered that only a loading dose was administered on day 1 in such patients, and two-point measurement is strongly recommended when TDM was planned in the morning on day 2.4)When only the trough concentration was measured on day 2 in non-ICU patients, the third dose may be postponed to optimise the dose until the confirmation of TDM results.5)Delaying TDM until near steady state (i.e., 3 days after vancomycin therapy) might be applicable in patients with mild/moderate MRSA infections and those without a risk of AKI (III-A).6)Performing earlier and frequent TDM is prudent in critically ill ICU patients (I).


*CQ 6. What is the target AUC in TDM?*


1)The target AUC/MIC for improving the efficacy of treatment for MRSA infection is ≥400 when using the MIC determined by the broth microdilution method (MIC_BMC_) [3,11,17,23,27,45,46,47,103,110,142,143,144,145,146,147] (I). The ratio should be ≥200 when using the MIC determined by Etest (MIC_Etest_) [148,149,150] (III-A).2)For empirical therapy before MIC determination, the target AUC should be ≥400 μg·h/mL presuming an MIC of 1 μg/mL [16,102,145,146,147,151,152,153,154,155,156,157,158] (III-A).3)Because of the insignificant difference of a 2-fold dilution in the measurement of MIC, the AUC/MIC ratio has excessive sensitivity to errors of MIC. Hence, the committee recommend the same AUC targets irrespective of the MIC even after determination of the MIC (III-A).4)To reduce the risk of AKI, the AUC should be ≤600 μg·h/mL [17,20,21,48,49,50,51,109,159,160,161,162,163,164,165,166,167,168] (I).5)In the actual clinical setting, the recommended target AUC is 400–600 μg·h/mL [17,110] (I).6)Because limited data are available regarding AUC targets for increasing treatment success rates against infections caused by antibiotic-resistant organisms other than MRSA [149,169,170,171,172,173,174] (III-B), revision of the dosing regimen should be decided according to the treatment response even at AUC/MIC < 400 for such infections.


*CQ 7. Is a continuous infusion administration strategy superior to intermittent infusion?*


1)A lower AKI risk was demonstrated with continuous administration than with intermittent administration using trough-guided dosing [175,176] (II).2)As the common practice of continuous infusion, after a loading dose (15–20 mg/kg), a maintenance dose (30–40 mg/kg) was administered continuously over 24 h (III-A).3)The PK level to monitor is the plateau level (i.e., steady-state concentration), and the target concentration is 20–25 μg/mL for continuous infusion (III-A).4)Because no comparative study on efficacy and safety between continuous and intermittent infusion therapy based on AUC-guided dosing is available, vancomycin continuous infusion is not currently recommended [175,176,177,178,179,180,181,182,183] (III-B).


*CQ 8. Is a loading dose required to achieve the target concentration and improve treatment efficacy?*


1)To increase the concentration on days 1–2, the routine use of a loading dose is required irrespective of renal function (I). This may increase the early treatment response (III-A).2)A loading dose does not increase the risk of AKI [55,184,185,186] (III-A).3)Not only the initial loading dose but also the subsequent maintenance doses have a significant impact on the achievement of AUC targets at steady state [55,185] (III-C).


*CQ 9. How can the dosage regimen be optimised to achieve the target AUC?*


1)Initial regimen
a)It is recommended to use a loading dose of 30 mg/kg (actual body weight) [55,184,185,186,187,188,189,190,191,192,193,194] (II). However, there are few safety-related data regarding loading doses of >3 g [193].b)It is recommended to use a maintenance dose of 20 mg/kg (actual body weight) q12h [194] (II). Careful use of daily doses of >4 g should be considered to prevent adverse effects [22,195].c)When increased vancomycin clearance is presumed (e.g., eGFR ≥ 130 mL/min/1.73 m^2^), maintenance doses of 15–20 mg/kg (actual body weight) q8h should be considered [196] (III-A).d)It is recommended to administer vancomycin for >1 h to prevent infusion-related reactions (II). Further prolonged administration (30 min/0.5 g) should be considered when doses of >1 g are used [197,198,199].e)The recommended dose can be modified using population PK models to achieve AUC targets assuming MIC = 1 μg/mL [72,200] (III-A).f)AUC on day 2 might be a better PK parameter than AUC at steady state to prevent adverse effects (III-A).2)Optimisation of the regimen based on the result of TDM
a)When drug concentrations (trough only or both peak and trough levels) are obtained, software predicts a dosing regimen that maximizes the likelihood of meeting the AUC target of 400–600 µg·h/mL for individual patients [17,54]. (II)b)In dose optimisation based on TDM results, a margin of error of ±20% should be considered in Bayesian estimation using population PK models derived from limited sampling (III-A). Therefore, it is recommended that the dose is adjusted to target an AUC of 500 µg·h/mL (margin of error = ±100).


*CQ 10. How can the dosage regimen be optimised to achieve the AUC targets in patients with impaired kidney function?*


1)Initial regimen
a)A nomogram for the tentative regimen was suggested according to body weight and CL_cr_ in patients with impaired kidney function (Table 2) (III-A).b)The tentative regimen should be individualized using the mean population PK model by entering patient data (gender, age, body weight and serum creatinine) into the software (III-A). A larger margin of bias should be considered for AUC estimated by the mean PK population model than for AUC estimated by the Bayesian method using individual patient serum concentrations. Hence, decreasing the upper threshold of AUC targets (i.e., 400–500 µg·h/mL) is suggested for the initial regimen to prevent overdose (III-A).2)Optimisation of the regimen based on the TDM results
a)Using Bayesian estimation, the regimen is optimised to achieve AUC = 400–600 µg·h/mL (II). However, bias even with Bayesian estimation should be considered to prevent overdose in patients with impaired kidney function [51,143,144,201,202,203,204,205,206,207,208,209,210,211,212].

### 3.2. Studies Conducted by the Committee

#### 3.2.1. CQ 1: Development of MIPD Software

This guideline committee developed MIPD software to promote AUC-guided dosing of vancomycin (termed PAT), in which redundant functions are thoroughly removed [53]. The feature of this web application, which is available on mobile phones, is programmed by R (https://www.r-project.org/, accessed on 12 January 2022), and it can display AUC both at steady state and on days 1 and 2 on the initial screen. The population PK model using peak–trough sampling in infected Japanese patients (*n* = 190) is incorporated for the Bayesian prior information [72]. The population covered the age range of 19.3–89.6 (mean 64.3) years, body weight range of 25.5–75.0 (52.3) kg, creatinine clearance range of 6.85–85.0 (77.1) mL/min, and volume of distribution (V_d_) range of 53.9–67.5 (60.7) L. Hence, paediatric patients, obese patients, patients with augmented renal clearance, and patients with sepsis/septic shock in whom V_d_ is increased substantially are not suitable for analysis by this software.

Oda et al. reported an external evaluation study and the ratios of Bayesian posterior AUC using trough sampling or peak–trough sampling to AUC using rich sampling based on the linear-up log-down trapezoidal method (AUC_REF_) were 0.93 (probability in the acceptable range of 0.8–1.2: 82.3%) and 0.95 (69.8%), respectively, in a population PK model developed using peak–trough sampling [53,72]. This result accorded with those of other reports [66,67,68,69,70,71] suggesting that a population PK model based on a large population using limited sampling such as routine TDM data is similarly applicable as that based on a small population with rich sampling in estimating Bayesian posterior AUC.

#### 3.2.2. CQ 2: Systematic Review and Meta-Analysis for Target Trough Levels and Comparison of Clinical Outcomes between AUC-Guided and Trough-Guided Dosing

a)
*The target range of trough levels*


Regarding the trough level, the treatment failure rates were significantly lower at ≥15 μg/mL than at <15 μg/mL (odds ratio [OR] = 0.63, 95% CI = 0.47–0.85, *p* = 0.003) [17]. Only two studies that compared treatment failure between trough levels of ≥10 and <10 μg/mL were available [22,23]. Although Kullar et al. demonstrated a significantly lower failure rate at ≥10 μg/mL in a single-centre retrospective analysis, when the other study was included in a meta-analysis, no significant difference was demonstrated [22].

The incidence of AKI was compared among trough levels of <10, 10–15, 15–20, and >20 μg/mL, and the incidence was significantly higher at higher trough levels in any comparison [17]. Trough levels of 15–20 μg/mL, which increased the treatment efficacy, represented a significant risk factor for AKI compared to levels of 10–15 μg/mL (OR = 1.63, 95% CI = 1.16–2.27, *p* = 0.004) [17]. In conclusion, a target trough level that ensures both efficacy and safety could not be determined.

b)
*AKI risk and treatment efficacy between AUC-guided dosing and trough-guided dosing*


AUC-guided dosing tended to more strongly lower the risk of AKI than trough-guided TDM (OR = 0.54, 95% CI = 0.28–1.01, *p* = 0.05) [17]. A meta-analysis of treatment efficacy could not be conducted because only one article was available. In conclusion, AUC-guided dosing for vancomycin was recommended because of its possibly superior safety profile.

#### 3.2.3. CQ 3: Candidates for AUC-Guided Dosing

To identify candidates for AUC-guided dosing, the committee conducted a multicentre study to demonstrate the risk factors for AKI during vancomycin therapy and to determine the threshold of trough levels that prevent AKI in the special population at high risk [52]. Independent risk factors for AKI were impaired kidney function (eGFR < 30 mL/min/1.73 m^2^) before treatment initiation, concomitant use of PIPC/TAZ or diuretics, trough levels of >20 μg/mL, and ICU stay. The incidence of AKI was 9.8%, and the cut-off trough level for AKI was 19.3 μg/mL in patients who received vancomycin therapy. To decrease the AKI rate to the level observed in all patients, the trough level had to be reduced to 12.4 μg/mL in patients with impaired kidney function, 13.5 μg/mL in those with concomitant use of PIPC/TAZ, and 11.7 μg/mL in those with the concomitant use of diuretics. Because a trough level of ≥15 μg/mL is required for successful treatment [17], trough-guided dosing cannot ensure safety in patients at high risk for AKI. Although no significant cut-off could be identified in ICU patients because of the small sample size, AUC-guided dosing might be mandatory for preventing AKI for such high-risk patients.

#### 3.2.4. CQ 4: AUC Estimation Using Trough-Only Measurement

When a population PK model based on rich sampling is used as a Bayesian prior, trough-only data can be used to generate accurate AUC estimates [66]. The committee studied the performance of AUC estimation with a population PK model based on limited sampling compared with the reference AUC calculated using the log-linear trapezoidal rule [53]. AUC estimation using trough and peak levels produced the least bias in patients treated with vancomycin q12h. Conversely, AUC estimation using only the trough level produced moderate and strong bias in patients treated with vancomycin q12h and q24h, respectively.

The committee also evaluated vancomycin AUC estimation using trough-only measurement [54]. The discrimination ability for early clinical outcomes using AUC cut-offs on day 2 (400 µg·h/mL for treatment response and 600 µg·h/mL for AKI) was confirmed only in patients who received q12 administration. In addition, a significant difference in early treatment response using the 400 µg·h/mL cut-off was obtained only in patients with low-risk MRSA infections (37/73 [50.7%] vs. 62/82 [75.6%], *p* = 0.001). Considering these results, we suspected that AUC estimation using only the trough concentration might be avoided in patients with difficult-to-treat MRSA infections and in patients with renal dysfunction who are likely to be prescribed once-daily dosing. AUC estimation using trough and peak levels is recommended in such patient populations.

#### 3.2.5. CQ 5: The Usefulness of AUC on Day 2

The committee confirmed that day 2 AUC ≥ 400 µg·h/mL was an independent factor for better early clinical response 48–72 h after the start of therapy (adjusted OR = 2.02, 95% CI = 1.15–3.53, *p* = 0.014), and day 2 AUC ≥ 600 µg·h/mL was an independent factor increasing the early occurrence of AKI 48–72 h after the start of therapy (adjusted OR = 44.77, 95% CI = 6.65–301.65, *p* < 0.001) [54]. The usefulness of AUC on day 2 was also described in CQs 1, 4, and 10.

#### 3.2.6. CQ 6: Systematic Review and Meta-Analyses for AUC/MIC Targets

A retrospective analysis was conducted in adult patients with MRSA bacteraemia for the association between AUC/MIC (cut-off of 400) and treatment failure [23,27,45,46,47] and for the association between AUC (cut-off of 600 μg·h/mL) and AKI [20,48,49,50,51]. A decreased risk of treatment failure was demonstrated in patients with high AUC/MIC (≥400) compared to that in patients with low AUC/MIC (<400, OR = 0.28, 95% CI = 0.18–0.45, *p* < 0.001) [17]. The safety analysis revealed that high AUCs (>600 μg·h/mL) significantly increased the risk of AKI versus low AUCs (≤600 μg·h/mL, OR = 2.10, 95% CI = 1.13–3.89, *p* = 0.02) [17]. From these results, the committee recommended an AUC target of 400–600 μg·h/mL presuming MIC = 1 μg/mL.

#### 3.2.7. CQ 8: Safety and Early Treatment Response of a Loading Dose

Ueda et al. reported no significant difference in the trough level at steady state irrespective of the use of a loading dose in patients with normal renal function (10.4 μg/mL vs. 10.2 μg/mL), and a loading dose did not result in higher incidence of nephrotoxicity compared to that for the conventional regimen (3.6% vs. 1.4% on the third day) [55]. In addition, a loading dose was associated with increased early clinical response rates 48–72 h after the start of therapy (OR = 4.59, 95% CI = 1.37–15.33, *p* = 0.013). The committee also conducted a large, multicentre retrospective study and confirmed that a loading dose was not a significant risk factor for AKI [52].

#### 3.2.8. CQ 9: Recommended Dosing Regimen

The committee demonstrated the AUCs on days 1 and 2 and at steady state according to four different regimens, all of which were recommended by the revised American guideline by IDSA and other organizations (maintenance dose of 15 or 20 mg/kg q12h with and without a loading dose [25 or 30 mg/kg], Table 3). With a loading dose, a higher day 1 AUC was obtained than achieved using the regimen without a loading dose, and a loading dose of 30 mg/kg might be required to achieve AUC targets. Both the loading and maintenance doses affect day 2 AUC, and regimen B-2 (a loading dose of 30 mg/kg followed by a maintenance dose of 20 mg/kg) appeared to be better than the regimens for achieving the target range, and this regimen was recommended as the initial therapy in this guideline. Compared to the median AUC at steady state of 471 µg·h/mL (interquartile range [IQR] = 374.7–556.8), the median trough level at steady state calculated using the actual interval from the pre-dose was 11.3 μg/mL (IQR = 8.1–13.7) in regimen B-2, suggesting that unnecessary dose increases would occur if trough-guided dosing is used.

The proportions of patients with day 2 AUCs of <400, 400–600, and >600 µg·h/mL with regimen B-2 were 29.6%, 56.3%, and 14.1%, respectively. Considering that only half of patients achieved the AUC targets with the recommended regimen, TDM on day 2 is recommended to optimise the dose in patients with serious or complicated MRSA infections.

#### 3.2.9. Nomogram of Vancomycin Dosing to Achieve AUC Targets

Considering an environment in which Bayesian software cannot be used, the committee proposed a nomogram of vancomycin dosing according to renal function for achieving an AUC of approximately 400–500 µg·h/mL on day 2 using PAT, which was developed by the committee (Table 3) [53]. A fixed maintenance dose was presented irrespective of body weight because CL_cr_, which was not adjusted for a standard body surface area of 1.73 m^2^_,_ was used in the nomogram. However, a loading dose was determined according to body weight (25–30 mg/kg) to prevent overdose in lean patients.

## 4. Discussion

### 4.1. Literature Review

#### 4.1.1. CQ 1. How Can MIPD Software Be Used to Increase the Accuracy of Dose Individualization?

Approximately 90 population PK models have been reported in the last two decades for vancomycin [61,62,63]. Recent debates have focused on better prior probability in a population PK model based on rich or limited sampling. Neely et al. reported an internal evaluation study in which the ratio of Bayesian posterior AUC using peak–trough sampling combined with a population PK model developed using peak–trough sampling (AUC_PT-PT_) to that using rich sampling combined with a population PK model developed using richly sampling (AUC_F-F_) was 0.86 (95% confidence interval [CI] = 0.81–0.93) [66].

Although the majority are used for research purposes, some software programs have been utilized in clinical settings to date. Turner et al. evaluated the ratios of Bayesian posterior AUC using peak–trough sampling to AUC_REF_ in five software packages [86]. The programs produced average accuracy ratios of 0.80 or higher and bias of less than 20% using the trough concentration alone for calculation. However, some programs had advantages of better accuracy or less bias, and some had characteristics of easier adaptation and use than other programs. 

Blouin et al. [96] showed a significant difference in weight-indexed V_d_ between obese and non-obese patients. Ducharme et al. [97] found that mean weight-indexed vancomycin V_d_ decreased with increasing body size. Hence, Bayesian posterior AUC for obese patients is required [71]. The ratio of the Bayesian posterior AUC using trough sampling to AUC_F-F_ ranged 0.87 (95% CI = 0.77–0.97) to 1.30 (95% CI = 1.19–1.40) among the three population PK models developed in nonspecific to obese patients. Colin et al. reported root mean square errors (as the index for predictive performance, smaller is better) for concentrations measured using two general population PK models of 2.44 and 4.18 μg/mL, respectively, whereas that measured using four obese population PK models ranged from 3.06 to 3.64 μg/mL among morbidly obese patients [73]. Taken together, although some population PK models have been reported for obese patients [61,63,71], a general population PK model may be used for Bayesian posterior AUC estimation in obese patients. 

Because large variable PK is observed in paediatric patients, these patients might benefit from MIPD in AUC-guided dosing. However, no direct evaluation study of the predictive performance of Bayesian posterior AUC using limited sampling in comparison to rich sampling has been reported to date. Harn et al. reported an external evaluation study in 13 paediatric patients [75,76]. The bias and precision for random/trough were −0.27 and 2.16 μg/mL, respectively, whereas those in the original study were 0.45 and 2.01 μg/mL, respectively. Berthaud et al. reported successful dose titration using Bayesian prediction for continuous dosing in paediatric patients in an RCT [77]. Target (AUC/MIC ≥ 400 and AUC ≤ 800 μg·h/mL) attainment rates at 24 h post-infusion for the Bayesian and control groups were 34/40 (85%) and 24/42 (57%), respectively (*p* = 0.007). Recently, Smit et al. reported a population PK model based on the largest paediatric population to date (population size = 1892), including overweight (247, 13%) and obese subjects (301, 16%) [78]. Han et al. summarized six programs for estimating Bayesian posterior AUC in paediatric patients [86]. Whereas the programs can adopt any paediatric model, one of the programs did not adopt a neonatal model. 

The predictive performance of Bayesian posterior AUC is limited in patients receiving RRT, although the Bayesian prior information has accumulating [61,79,80,81,82,83]. Oda et al. reported an external evaluation study of a population PK model for patients receiving continuous RRT (CRRT) [82]. The target (trough concentration of 10–20 μg/mL) attainment rate in patients who underwent dose titration using Bayesian prediction was 87.0% (20/23), which was significantly higher than that of 53.8% (7/13, *p* = 0.046) achieved using traditional TDM (using creatinine clearance). The software based on Visual Basic for Applications was developed for patients receiving CRRT [82]. 

Turner et al. reported that Bayesian estimation using a single non-trough sample produced similar results as that using trough sampling among most software programs [97]. Ueda et al. found that the actual recorded sampling times after the dose were <10 (5.9%), 10–11 (22.8%), 11–12 (55.0%), and ≥12 h (16.3%) in patients treated with vancomycin q12h and were <22 (24.1%), 22–23 (27.6%), 23–24 (41.4%), and >24 h (6.9%) in patients treated with vancomycin q24h [24]. If these were assumed equally as trough levels, this might cause a substantial error in the evaluation of dose optimisation. Neely et al. reported that fewer than half of the samples were within the trough concentration window of 10–12 h post-dose. Because of the poor adherence to hospital policy, an optimal sampling strategy rather than a trough concentration-based sampling strategy was introduced, and they achieved significantly tighter control of AUCs around the targets using optimal sampling [18].

#### 4.1.2. CQ 2. What Are the Recommended PK/PD Parameters for TDM?

PK/PD analysis using a neutropaenic mouse thigh infection model demonstrated that the AUC/MIC ratio most strongly correlated with bactericidal effects against methicillin-susceptible *Staphylococcus aureus* and MRSA [102]. A meta-analysis in the actual clinical setting revealed that higher AUC/MIC was significantly correlated with lower mortality and treatment failure rates [103]. Therefore, AUC/MIC should be employed to predict clinical and bacteriological efficacy [3,11]. In vitro studies with low-susceptibility strains demonstrated the correlation between AUC and the incidence of heterogeneous vancomycin-intermediate *Staphylococcus aureus* (hetero VISA) [104,105,106].

The occurrence of AKI was investigated in rats, suggesting that urinary KIM-1 levels can be used for the early detection of VCM-induced AKI [107]. Increased urinary KIM-1 levels were more strongly correlated with AUC than with trough levels [108]. Aljefri et al. conducted a meta-analysis of clinical studies, demonstrating that higher AUC was significantly correlated with a higher incidence of AKI [109]. Therefore, AUC should be employed to predict the risk of AKI [3]. Basic and clinical studies suggested that AUC is the most appropriate PK/PD parameter indicating the efficacy and safety of vancomycin. In the actual clinical setting, trough levels have been substituted for AUC [11,110]. The relationship between trough levels and AUC has recently been reviewed, and AUC could not be correctly evaluated using trough levels [54,56,66,111,112,113]. Rees et al. reported that the incidence of AKI was significantly lower with AUC-guided dosing (5.7%) than with trough-guided dosing (23.1%), and the treatment failure rate was 15.1% and 24.6%, respectively [114]. The occurrence of AKI is related to longer hospital stays, additional treatment costs, and increased mortality [18,115]. Lee et al. reported that AUC-guided dosing minimizes the risk of AKI, thereby reducing treatment costs [116].

#### 4.1.3. CQ 3. Who Are the Candidates for AUC-Guided Dosing?

AUC-guided dosing is recommended for patients with risk factors for AKI during vancomycin treatment such as prolonged treatment [117,118], impaired kidney function [117,118,119], the concomitant use of nephrotoxic drugs (aminoglycosides, amphotericin B, contrast medium, and frequent use of non-steroidal anti-inflammatory drug [NSAIDs]) and PIPC/TAZ [6,118,120,121,122,123], the use of hypertensive drugs [117], dehydration [117,119], and severe illness [118].

Patients with altered V_d_ might be candidates for AUC-guided dosing to increase the prediction accuracy [42,52,117,118,119,137]. V_d_ was 0.4 L/kg [96] in healthy adults, 0.7 L/kg [125] in patients with infectious diseases, and 1.3 L/kg [126] in patients with septic shock. Augmented renal clearance [127] is experienced by patients with sepsis [128,129,130]. Enhanced clearance is also observed in patients with hematopoietic tumours or febrile neutropaenia [131]. Furthermore, patients with heart failure [132,133], oedema [132,134], dehydration [117,119], burns [135,136], obesity [137,138,139,140], or emaciation [140] have different V_d_ and clearance from individuals in the normal population.

#### 4.1.4. CQ 4. Is the Trough Concentration Alone Sufficient to Estimate AUC Using Bayesian Software?

Although it is preferred to obtain two PK samples to accurately estimate AUC using the Bayesian approach, updated TDM for vancomycin guidelines suggested that the trough level alone may be sufficient to estimate AUC in some patients [3]. Utilizing only the trough concentration, accuracy (range 0.79–1.03) and bias (range 5.1–21.2%) were reported using Bayesian dose-optimising software [98]. Although variation was present, the achievement of therapeutic PK targets was substantially higher for the AUC estimated using the Bayesian method with trough-only measurements than using traditional trough monitoring without the Bayesian approach. Neely et al. reported that AUC ≥ 400 µg·h/mL indicated trough concentrations of <15 μg/mL in 68% of cases [18]. Using data from the trough-only measurement, Ueda et al. reported that the median trough level was only 11.0 μg/mL for AUC = 400–600 µg·h/mL [54].

#### 4.1.5. CQ 5. When Should TDM Be Performed?

Previously, it was recommended to perform initial TDM immediately prior to the fourth or fifth dose including the loading dose on day 3 when steady state was reached. Takahashi et al. reported that steady-state VCM serum concentrations were not achieved on day 3 in patients with impaired kidney function, and underestimation of the trough level on day 3 should be considered in those patients [141]. To improve clinical outcomes, early achievement of the target vancomycin concentration is preferable. Because Bayesian estimation does not require steady-state serum vancomycin concentrations, it enables the early assessment of AUC target attainment. Casapao et al. reported that higher day 1 exposure resulted in lower rates of clinical failure and persistent bacteraemia in patients with MRSA bacteraemia [26]. However, in a prospective, multicentre study of adult patients with MRSA bacteraemia, a higher day 2 AUC/MIC ratio was not associated with a lower rate of failure, but it was associated with the risk of AKI [51]. The authors concluded that day 2 AUCs should be maintained at less than approximately 515 µg·h/mL to minimize the likelihood of AKI.

#### 4.1.6. CQ 6. What Is the Target AUC in TDM?

Prybylski [142] conducted a meta-analysis demonstrating a significantly reduced treatment failure rate at a high AUC/MIC in patients with *S. aureus* bacteraemia in 2015. Subsequently, Men et al. [103] conducted a systematic review suggesting an AUC/MIC ratio of 400 as a threshold for efficacy. By contrast, Casapao et al. [143] reported that AUC/MIC of 600 on the first day as the cut-off for treatment efficacy, and Lodise et al. [144] reported values of 521 and 650 as cut-offs for efficacy on the first and second days, respectively. However, they mentioned that AUC determined in their institution was equivalent to a value of approximately 400 in other papers. In conclusion, AUC/MIC ≥ 400 (determined by the broth microdilution method) serves as a predictor of efficacy against MRSA (MIC ≤ 1 μg/mL) infection [3,11,110]. As MICs determined by Etest are higher than those by the broth microdilution method, a threshold of 212 has been reported for AUC/MIC (determined by Etest) [148]. Holmes et al. suggested that AUC/MIC of 400 by the broth microdilution method is equivalent to 226 by Etest [149].

The MIC of VCM is generally 1 μg/mL [145,146,147]. The MIC distribution of MRSA strains obtained from the European Committee on Antimicrobial Susceptibility Testing MIC distribution website (https://mic.eucast.org/Eucast2/; accessed on 12 February 2022) was 12.7% in MIC at 2 μg/mL, 83.1% in MIC at 1 μg/mL, and 4.1% in MIC at 0.5 μg/mL. The target AUC for treatment success to achieve AUC/MIC ≥400 is ≥800 µg·h/mL, ≥400 µg·h/mL, and ≥200 µg·h/mL, respectively. For an MIC of 2 μg/mL, the conventional dosing regimens failed to achieve the targeted AUC/MIC ratio. In the treatment of MRSA infections with vancomycin MIC at 2 μg/mL, determination of the dose producing AUC/MIC ≥ 400 is impractical and might cause adverse effects [161,162,163,164,165,166,167,168]. For an MIC of 0.5 μg/mL, there are no data supporting dose reductions to achieve an AUC/MIC ≥ 400 (AUC ≥ 200 µg·h/mL) and the emergence of hetero VISA was associated with an AUC of <400 μg·h/mL [104,105,106]. Considering the limited accuracy of automated susceptibility testing methods, we recommended a target AUC of ≥400 µg·h/mL irrespective of MIC. 

The MIC of VCM is generally 1 μg/mL [145,146,147]. The MIC distribution of MRSA strains obtained from the European Committee on Antimicrobial Susceptibility Testing MIC distribution website (https://mic.eucast.org/Eucast2/; accessed on 12 February 2022) was 12.7% in MIC at 2 μg/mL, 83.1% in MIC at 1 μg/mL, and 4.1% in MIC at 0.5 μg/mL. The target AUC for treatment success to achieve AUC/MIC ≥400 is ≥800 µg·h/mL, ≥400 µg·h/mL, and ≥200 µg·h/mL, respectively. For an MIC of 2 μg/mL, the conventional dosing regimens failed to achieve the targeted AUC/MIC ratio. In the treatment of MRSA infections with vancomycin MIC at 2 μg/mL, determination of the dose producing AUC/MIC ≥ 400 is impractical and might cause adverse effects [161,162,163,164,165,166,167,168]. For an MIC of 0.5 μg/mL, there are no data supporting dose reductions to achieve an AUC/MIC ≥400 (AUC ≥200 µg·h/mL) and the emergence of hetero VISA was associated with an AUC of <400 μg·h/mL [104,105,106]. Considering the limited accuracy of automated susceptibility testing methods, we recommended a target AUC of ≥400 µg·h/mL irrespective of MIC. 

Aljefri et al. [109] conducted a systematic review suggesting that AUC of 650 μg·h/mL is a threshold for preventing AKI. Suzuki et al. [159] found that AUC was approximately 600–800 in patients with AKI and 400–600 μg·h/mL in patients without AKI (*p* = 0.014). In addition, they observed that the rates of AKI in patients with AUCs of 400, 600, and 800 μg·h/mL were 6.2%, 12.9%, and 24.7%, respectively [160]. The guideline committee recommended a target AUC of 400–600 μg·h/mL; however, the evidence regarding the AUC/MIC threshold for complicated infections is limited. Only one retrospective study assessed the treatment of osteomyelitis [149], in which a cut-off of 293 was determined for AUC/MIC using Etest (more than half of strains had MIC > 1 μg/mL).

Few reports have been published on the target AUC/MIC for vancomycin against infections caused by bacteria other than *S. aureus*. An in vivo experiment revealed an approximately 10-fold difference in AUC/MIC for bactericidal effects (399 for *S. aureus* and 39.1 for *Streptococcus pneumoniae*) [169]. Jumah et al. [170] reported a significantly lower mortality rate at AUC/MIC_Etest_ ≥ 389 in patients with bacteraemia caused by enterococci. Although the guidelines recommended AUC/MIC_Etest_ ≥ 200 for MRSA, the higher value of MIC_Etest_ for enterococci [170] than for MRSA [149] should be considered. Different target AUC/MIC ratios according to bacterial species have been reported for other anti-MRSA drugs. The bactericidal effects (1 log killing) of daptomycin against *S. aureus*, *S. pneumoniae*, and *Enterococcus faecium* were observed with AUC/MIC ratios of 666 ± 87, 290 ± 121, and 4.14–33.8, respectively [171]. The bacteriostatic effects of linezolid against *S. aureus* and *S. pneumoniae* were achieved at AUC/MIC ratios of 82.9 ± 57 and 48.3 ± 29, respectively [172]. Matsumoto et al. considered that treatment success was more likely at AUC/MIC > 900 μg·h/mL for teicoplanin [173]. Because the teicoplanin MIC for coagulase-negative staphylococci (CNS) is higher than that of MRSA [174], the applicability of the target PK/PD parameter to CNS remains unclear.

Only the unbound fraction of a drug is pharmacologically active, and is responsible for antimicrobial activity and can cause toxicity. Hence, free AUC/MIC target ≥200 has been advocated as the PK/PD target assuming a fixed unbound vancomycin fraction of 50% [151]. The total concentration of high protein-binding drugs such as teicoplanin was decreased in patients with hypoalbuminemia [152]. Vancomycin is generally considered a moderately protein-bound antibiotic (unbound fraction rate: 45.4–72.9%) [153,154,155]. A significant correlation between the unbound fraction and the albumin concentration was reported for vancomycin by several authors [154,156,157]. Critically ill patients exhibit marked variability in serum albumin concentrations, which may alter the protein binding. De Cock et al. [151] reported that unbound vancomycin concentrations in paediatric patients were adequately predicted using the following equation: unbound vancomycin concentration (mg/L) = 5.38 + [0.71 × total vancomycin concentration (mg/L)] − [0.085 × total protein concentration (g/L)]. A 1–8-fold increase in the vancomycin MIC was reported, possibly because of the decrease in unbound concentration as a result of the presence of albumin in the broth [102]. Because routine monitoring of the unbound concentration is not feasible in clinical practice, the target total concentration might be lowered depending on the degree of hypoalbuminemia in teicoplanin [16,158]. However, no recommendation for the desired vancomycin AUC was made in patients with severe hypo- or hyperalbuminemia in this guideline because of the unavailability of the outcome result from a clinical trial.

#### 4.1.7. CQ 7. Is a Continuous Infusion Administration Strategy Superior to Intermittent Infusion?

Bissell et al. reported that the continuously infused vancomycin had a shorter time to target achievement with a higher incidence of target attainment than intermittently infused vancomycin in critically ill patients, which resulted in a lower average number of blood samples per patient and shorter duration of therapy [181]. A recent meta-analysis of RCTs and observational studies for critically ill adult patients reported by Flannery et al. demonstrated that continuous infusion was associated with a lower AKI risk (OR = 0.47, 95% CI = 0.34–0.65) and a higher PK target attainment rate (OR = 2.63, 95% CI = 1.52–4.57) than intermittent infusion [176]. The meta-analysis conducted by Chu et al. [175] also demonstrated a low AKI risk for continuous infusion. Although most continuous infusion studies did not report vasculitis, Vuagnat et al. reported two cases of catheter-associated phlebitis among 23 patients who underwent continuous infusion therapy [182].

The evaluated PK variable was the trough level for intermittent infusion and the plateau level (i.e., steady-state concentration) for continuous infusion [175,176]. As the most common practice, the maintenance dose (e.g., 30 mg/kg) was administered continuously over 24 h after a loading dose (15–20 mg/kg), and the plateau concentration was sustained after reaching steady state with continuous infusion [179]. Therefore, the PK target is not the trough concentration, but instead, it is the plateau concentration in an continuous infusion strategy. The target plateau concentration was 20–25 μg/mL for continuous infusion and trough-guided dosing (15–20 mg/L) was conducted for intermittent dosing. The average time to achieve a vancomycin trough level of 15 μg/mL with intermittent infusion was 50 ± 21 h, versus 16 ± 8 h to reach a serum plateau level of 20 μg/mL with continuous infusion (*p* < 0.001).

Continuous administration simplifies TDM through easier AUC estimation than intermittent administration, in which AUC was estimated using Bayesian software. Compared to trough concentrations in intermittent administration, the plateau concentration with continuous infusion is a more trustworthy surrogate of AUC. Plateau concentrations of 17.5–22.5 μg/mL correspond to AUCs of 420–540 μg·h/mL [183]. Once the infusion is started, a serum concentration at one point is measured after 24–48 h, which is likely to represent the steady-state value in patients with normal renal function. The daily dose or hourly dosing rate can then be optimised proportionately to achieve the desired plateau concentration.

Considering that a higher dose tends to be required for trough-guided dosing than for AUC-guided dosing, it is expected that smaller total daily doses can be used to achieve target serum concentrations, resulting in a lower AUC and lower risk of AKI with continuous administration than with intermittent administration. Hutschala et al. calculated the AUCs for continuous and intermittent infusions as 529 ± 98 and 612 ± 213 μg·h/mL, respectively [177]. Similarly, Wysocki et al. estimated these values as 577 ± 120 and 653 ± 232 μg·h/mL, respectively [178]. In addition, the AUCs exhibited significantly less variation for continuous infusion than for intermittent infusion [178]. Hence, using the same PK targets (AUC = 400–600 μg·h/mL), a comparative study with intermittent administration based on AUC-guided dosing should be performed to demonstrate the beneficial effect of continuous infusion. Recently, Garreau et al. conducted a population PK analysis and dosing simulation targeting AUCs of 400–600 μg·h/mL on day 2 in critically ill patients who underwent continuous infusion therapy [180].

#### 4.1.8. CQ 8. Is a Loading Dose Required to Achieve the Target Concentration and Improve Treatment Efficacy?

Starting doses commonly recommended for patients with sepsis/septic shock frequently fail to achieve the desired target exposures because of the change of V_d_ caused by extravasation. In general, a vancomycin loading dose of 20–35 mg/kg is suggested to rapidly attain the target concentration in critically ill patients [3]. However, if a loading dose increases treatment efficacy without increasing the risk for AKI, routine loading should be considered even in clinically stable patients.

Rosini et al. reported that the trough level 12 h after the start of vancomycin therapy allowed a significantly greater proportion of patients to achieve the target of 15 μg/mL using a loading dose of 30 mg/kg compared to the findings for patients who did not receive a loading dose [185]. However, the effect of the loading dose was attenuated gradually, and only 20% of patients with a loading dose attained the target trough level at steady state. A loading dose alone may not be sufficient to increase the trough level measured 48 h after the initial dose in patients with normal vancomycin clearance. The main purpose of the loading dose is not to obtain the target concentration at steady state, but instead, the goal is the rapid achievement of a therapeutic concentration within 12–24 h [186,187,188,189].

Casapao et al. evaluated the association between the day 1 vancomycin exposure profile and outcomes among patients with infective endocarditis associated with MRSA, and a lower AUC/MIC ratio was independently associated with treatment failure [143]. An initial daily dose of <40 mg/kg was a risk factor for vancomycin non-responsiveness in MRSA pneumonia [190]. An initial dose of >20 mg/kg led to a faster resolution of systematic inflammatory response syndrome [191]. Although additional research of the loading dose is required in patients with impaired kidney function, the committee recommended the routine use of a loading dose of vancomycin in the treatment of MRSA infections.

#### 4.1.9. CQ 9. How Can the Dosage Regimen Be Optimised to Achieve the Target AUC?

The revised IDSA guidelines recommended doses of 15–20 mg/kg every 8–12 h and suggested a loading dose of 20–35 mg/kg to rapidly achieve the target concentration in critically ill patients. The recommended vancomycin dosing regimens to achieve AUC/MIC > 400 were evaluated using PK/PD modelling and Monte Carlo simulations. All evaluated regimens resulted in target attainment probabilities (PTAs) of >90% only for MICs of 0.5 and 0.75 mg/L, and daily doses exceeding 3 g (i.e., 1.5 g q12h) were required for an MIC of 1 μg/mL in patients with cancer and an average body weight of 72.7 kg [201]. Alqahtani et al. conducted model-based evaluation of the standard dosing regimen in patients who underwent open-heart surgery [202]. Although the PTA of AUC/MIC > 400 for MIC = 1 μg/mL was less than 50% with the regimen of 15 mg/kg q12h, the value exceeded 80% for the regimen of 20 mg/kg q12h.

Excessive doses are required for patients with increased creatinine clearance (CL_cr_). Vancomycin doses of 2 g every 8 h in adult patients (median body weight = 75 kg; CL_cr_ = 107 mL/min/1.73 m^2^) with sepsis or septic shock were needed to achieve optimal therapeutic exposure (AUC ≥ 451 µg·h/mL) [203]. Tsai et al. recommended a regimen of 2 g every 8 h for patients with CL_cr_ > 131 mL/min/1.73 m^2^ [204].

#### 4.1.10. CQ 10. How Can the Dosage Regimen Be Optimised to Achieve the AUC Targets in Patients with Impaired Kidney Function?

Most nomograms for vancomycin dosing in patients with impaired kidney function were generated to achieve the targeted trough level [69,206,207,208,209]. Thomson et al. reported that the trough level of 10–20 μg/mL was achieved in 71% of patients using their nomogram [69]. Kullar et al. developed a nomogram for a target trough concentration of 15–20 μg/mL [207]. In total, 58% of patients achieved the initial target trough concentration, and only 4.5% of patients developed AKI. Elyasi et al. reviewed nomograms targeting high trough levels of vancomycin [210]. Although most of these nomograms significantly increased the achievement rate of target trough concentrations, they have only been validated in narrow groups of patients.

Few nomograms have been reported for obtaining an AUC of >400 μg·h/mL. Oda et al. develop a nomogram for vancomycin dosing to obtain the AUC target of 400 µg·h/mL, and therapeutic AUC and trough range attainment rates were 63.8% and 70.2%, respectively [194]. Lines et al. created a vancomycin dosing nomogram based on trough-only extrapolated AUC according to the dose range of 12.5–20 mg/kg and observed a trough level range of 8–19 μg/mL [211]. The extrapolated AUC dosing method had a significantly lower incidence of AKI than the conventional dosing method targeting trough levels of 15–20 μg/mL. The median AUC in the conventional dosing group was 617 µg·h/mL, which exceeds the AUC targets recommended in this guideline. Niinuma et al. investigated whether the loading dose affects safety in patients with impaired renal function (30 mL/min/1.73 m^2^ ≤ eGFR < 80 mL/min/1.73 m^2^) [212]. The incidence of AKI was not significantly different between patients receiving loading doses of 25 or <25 mg/kg (9.1% vs. 8.9%).

### 4.2. Limitation 

This review had several limitations. First, because of the unavailability of a population PK model for paediatric patients, obese patients, patients with augmented renal clearance, and patients with sepsis/septic shock in Japan, the committee could not report evidence concerning TDM in these special populations, and recommendations for such patients were not provided. Second, although the incidence of AKI tended to be lower with AUC-guided dosing than with trough-guided dosing, there was no significant difference (OR = 0.54, 95% CI = 0.28–1.01) in our meta-analysis [17]. If a recent RCT that reported a significantly lower AKI rate in the AUC-guided dosing group was included in the meta-analysis, the beneficial effect of this strategy might have been demonstrated [114]. Third, a recommendation for or against continuous infusion was not made. Research comparing safety or efficacy between continuous and intermittent infusion should be performed using AUC-guided dosing. Fourth, although a nomogram according to renal function was suggested using the software, verification is required for its utility. Hence, the proposed regimen should be adjusted using early TDM in patients with impaired kidney function. Finally, only observation studies were included in the meta-analysis that demonstrated the tendency of a lower AKI risk with AUC-guided dosing and the AUC target range to achieve.

## 5. Conclusions

The guideline provided recommendations for AUC-guided dosing for vancomycin. To expand the use of AUC-guided dosing, the availability of verified open and free Bayesian dose-optimizing software programs is required. In addition, the development of user-friendly programs that are adaptive for special patient populations is the next issue to be resolved in clinical practice. Although physicians should be fully responsible for their prescriptions, education and support by academia via the use of software are mandatory. 

## Figures and Tables

**Table 1 pharmaceutics-14-00489-t001:** Category for ranking recommendations and evidence levels adopted in the guidelines. Referenced from the report [16] developed by this committee, Oxford University Press, 2022.

Category, Grade	Definition
I	Strong recommendation with strong evidence for efficacy with clinical benefit
II	General recommendation with moderate evidence for efficacy with clinical benefit
III-A	Suggestion to encourage use by expert opinion without sufficient evidence
III-B	Insufficient evidence to make any suggestion
III-C	Suggestion to discourage use because of insufficient evidence
IV	Recommendation against use with sufficient evidence of no clinical efficacy or increased adverse outcomes

**Table 2 pharmaceutics-14-00489-t002:** Vancomycin nomogram according to renal function for the achievement of AUC of approximately 400–500 µg·h/mL on day 2.

CL_cr_ (mL/min)	Daily MD	80 kg		70 kg		60 kg		50 kg		40 kg	
		LD	AUC_24–48h_	LD	AUC_24–48h_	LD	AUC_24–48h_	LD	AUC_24–48h_	LD	AUC_24–48h_
100	1.25 g × 2	2 g	512	2 g	512	1.75 g	504				
90	1.0 g × 2	2 g	465	2 g	465	1.75 g	456	1.5 g	446		
80	1.0 g × 2	1.75 g	496	1.75 g	496	1.75 g	496	1.5 g	494		
70	0.75 g × 2	1.75 g	449	1.75 g	449	1.75 g	449	1.5 g	435	1.25 g	420
60	0.75 g × 2	1.75 g	511	1.75 g	511	1.75 g	511	1.5 g	492	1.25 g	474
50	0.5 g × 2	1.75 g	446	1.75 g	446	1.75 g	446	1.5 g	423	1.25 g	400
40	0.5 g × 2	1.75 g	528	1.5 g	497	1.5 g	497	1.5 g	497	1.25 g	467
30	0.75 g × 1	1.75 g	512	1.5 g	472	1.5 g	472	1.5 g	472	1.25 g	432

CL_cr_: creatinine clearance; MD: maintenance dose; LD: loading dose; AUC_24–48h_: area under the concentration–time curve on day 2.

**Table 3 pharmaceutics-14-00489-t003:** Distribution of AUC at initial TDM in each regimen for patients with normal renal function.

Regimen	LD	MD (q12h)	No. of Patients	Median (IQR) of AUC (μg·h/mL)		
				Day 1	Day 2	Steady State
A-1	None	15 mg/kg	69	321.9 * (265.1–396.1)	390.4 * (326.5–444.6)	417.9 * (354.5–477.0)
A-2	None	20 mg/kg	20	355.9 * (303.1–455.7)	429.8 (378.5–488.8)	456.1 (420.0–503.3)
B-1	25 mg/kg	15 mg/kg	64	410.0 * (352.3–473.5)	407.0 * (353.0–454.6)	422.3 ** (351.8–473.8)
B-2	30 mg/kg	20 mg/kg	71	472.2 (403.3–543.4)	459.9 (369.2–530.5)	472.0 (374.7–556.8)

AUC: area under the concentration-time curve; TDM: therapeutic drug monitoring; LD: loading dose; MD: maintenance dose; IQR: interquartile range. *: vs. B-2, *p* < 0.01; **: vs. B-2, *p* = 0.01.

## Data Availability

Not applicable.

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
