# Peer review of "Clinical Practice Guidelines for Therapeutic Drug Monitoring of Vancomycin in the Framework of Model-Informed Precision Dosing: A Consensus Review by the Japanese Society of Chemotherapy and the Japanese Society of Therapeutic Drug Monitoring"

_pharmaceutics, 2022, doi:10.3390/pharmaceutics14030489_

Round 1

Reviewer 1 Report

This paper is a consensus review of the updated clinical guidelines for therapeutic drug monitoring of vancomycin in adult patients. It has been prepared by the joint Japanese committees of chemotherapy and therapeutic drug monitoring in order to disseminate the latest updates on vancomycin dosage and monitoring for the optimised antimicrobial efficacy and safety.

The manuscript is well-written and structured according to very transparent rules. It is comprehensive and rooted in the existing scientific literature and clinical practices. Therefore, I would like to sincerely congratulate the authors their work.

I have only one formal concern. In L:162-164: the authors write "The Japanese version of the guidelines was approved by the JSC and JSTDM Board of Directors, and it will be published in Japanese Journal of Chemotherapy in February 2022." Since the current paper is actually a set of guidelines/a whitepaper concerning improved methods of vancomycin dosage and therapeutic monitoring, I wonder if the publication in the Japanese Journal of Chemotherapy (another scientific journal) will differ substantially enough to consider these publications as separate pieces of work.  I do not perceive language difference substantial enough. If they are the same, this may suggest autoplagiarism which, I believe, is not the authors' aim. I would like to ask the authors to address this issue. Perhaps publishing one of them as a "commentary" or "statement" would solve this?

Additionally, I would like to ask the authors to explain the issue of plasma protein binding of vancomycin. This may have some effect when translating the in vitro MIC assessment to the desired AUC values. The plasma protein binding of vancomycin is around 50%. Please comment on that.

There are also some minor editorial points:

L505: Association between (“of” to be removed)

L506: “decreased”, not “deceased”

L530: The abbreviation BMC in MIC(BMC) could be explained earlier for clarity.

L549: Writing S. pneumonia just after S. aureus may mislead the reader that these are the same genus. I suggest writing Streptococcus in full.

Tables: I suggest revising the table editing, particularly the lines dividing columns and verses.

Reviewer 2 Report

Authors presented the interesting and very useful clinical practice guidelines for therapeutic drug monitoring of vancomycin in adult patients. The recommendations provided by the Authors were  prepared based on the conducted studies and literature review. The conclusion regarding superiority of AUC-guided dosing over trough-guided dosing has pivotal meaning in clinical practice. Some issue need to be addressed before publication:

  1. Page 5, lines 187-188: “rich sampling” and “limited sampling” terms should be explained
  2. The guideline recommends the MIPD software to promote AUC-guided dosing. How the software can be accessed? Is it available for users from other countries apart from Japan?
  3. Page 8, line 349: CQ2 is doubled (section 3.2. and 3.3) but CQ3 is missing.
  4. Page 9, lines 422-425: if the trough concentration shouldn’t be used for the AUC estimation, how many concentration time points do you recommend in case of patients with difficult-to-treat MRSA infections and renal disfunction?
  5. Page 11, lines 485-487: Authors claim that AUC/MIC is sensitive to errors of MIC. What about errors related to vancomycin determination? Do you recommend any analytical method for TDM of vancomycin to avoid the errors?
  6. Page 16, line 728: should be “area under the concentration-time curve”
  7. Page 16, lines 756-758: the presented nomogram (Table 3) can be very useful for the units in which Bayesian software is unavailable. Have you confirmed the utility of the nomogram in patients? How many patients achieved the target AUCs using the nomogram?

Reviewer 3 Report

 Overall, the review should be streamlined, shortened and more clearly arranged. This would support better readability and comprehensibility.

Especially, the abstract should be completely revised. E.g. Background/Conclusion: Model-informed dosing should be promoted, but as a conclusion there are recommendations on AUC-guided dosing. I understand this is related to each other, but seems confusing for the reader. “The regimen likely to achieve AUC targets was suggested” – Is there only one? Is there something missing in the sentence? Language should be revised.

Further comments:

Page 2 Line 49:  infusion-related reaction is not dose depended and not improved by TDM. Please leave out the reaction in this context.

Page 2 Line 76: Please revise the aim of the review. Maybe two sentences would improve the understanding.

Page 6 Line 249: That predictive performance… Is there a relation to a before mentioned predictive performance? Why is there a new paragraph? If this is a new thought/ a new paragraph, please revise language for better readability.
